# Investigation of the Mechanical Properties of Iron Tailings Concrete Subjected to Dry–Wet Cycle and Negative Temperature

**DOI:** 10.3390/ma16134602

**Published:** 2023-06-26

**Authors:** Xiaozhou Liu, Hu Xu, Ben Li, Chen Zhang, Yu Zhang, Canhao Zhao, Kaihang Li

**Affiliations:** 1College of Civil Engineering and Architecture, Quzhou University, Quzhou 324000, China; 2Advanced and Sustainable Infrastructure Materials Group, School of Transportation, Civil Engineering and Architecture, Foshan University, Foshan 528000, China; 2112161010@stu.fosu.edu.cn (H.X.);

**Keywords:** iron tailings concrete, dry–wet cycling, negative temperature, mechanical performance, optimal substitution rate

## Abstract

This research investigates the effects of iron tailings content on the mechanical properties and durability of concrete under dry–wet cycling and negative temperature conditions (−10 °C), where iron tailings replace river sand at rates of 0%, 10%, 20%, and 30%. A variety of tests were conducted on the iron tailings concrete, including compressive strength, flexural strength, splitting tensile strength, mass loss, and relative dynamic modulus, and its pore characteristics were analyzed using low-field nuclear magnetic resonance (NMR) experiments. The results reveal that when 20% of the river sand was replaced with iron tailings, the concrete achieved optimal splitting strength, compressive strength, and flexural strength at 28 days, improving by 0.46 MPa, 3.14 MPa, and 0.41 MPa, respectively, compared to conventional concrete. Furthermore, the concrete containing this proportion of iron tailings demonstrated superior mechanical properties and durability in both negative temperature conditions and dry–wet cycling experiments. Due to the excellent physical and chemical properties of iron tailings, they enhance the performance of concrete when incorporated in appropriate quantities. The fine granularity of iron tailings helps to compensate for the granularity defects in concrete aggregates by filling internal voids, optimizing the pore structure, and improving the concrete’s density and integrity. This enhances the concrete’s mechanical properties and its resistance to external solutions and harmful ion penetration. Additionally, the active substances in iron tailings promote the hydration reaction of cement, leading to the formation of an increased amount of C-S-H gel and other hydration products in the cement system.

## 1. Introduction

Iron tailings are among the significant by-products of mining operations and a major component of industrial solid waste. As the steel industry develops, the proportion of iron tailings in industrial solid waste has been continuously increasing [1]. The annual production of iron tailings in China exceeds 265 million tons, but the utilization rate is only 6.95% [2]. Large-scale accumulation of iron tailings results in land resource waste and poses severe threats to the surrounding ecological environment [3,4,5,6,7,8]. Therefore, the treatment and comprehensive utilization of iron tailings carry substantial economic and environmental significance. It is essential to find an eco-friendly and cost-effective method for recycling and utilizing iron tailings, which will help alleviate the storage burden on mines and enhance the inherent value of iron tailings. The chemical composition of iron tailings primarily comprises aluminum, silicon, and calcium oxides, with calcium and silicon oxides making up over 50% of the total content. This characteristic offers potential for the application of iron tailings in the building materials industry [9]. However, the stable crystal structure within iron tailings results in relatively low reactivity [10,11], which limits their in-depth development and application, ultimately constraining their overall value. In concrete production, iron tailings are often employed as a fine aggregate substitute for river sand [12,13]. This approach effectively harnesses the advantageous physical properties of iron tailings, such as their small particle size and large specific surface area, as well as their potential pozzolanic activity, while also consuming a substantial amount of iron tailings.

Owing to the simple and energy-efficient production process of iron tailings concrete (ITC), it effectively addresses the environmental concerns related to iron tailings resources. In recent years, researchers from around the world have extensively studied the utilization of iron tailings in concrete, making strides in the preparation and performance testing of ITC. A multitude of studies suggest that although the excellent water absorption and substantial specific surface area of iron tailings may increase water usage in concrete and diminish its workability [14,15], the fine particle size and high hardness of iron tailings can enhance the mechanical properties of concrete. This renders iron tailings as excellent concrete admixtures and the optimal dosage range for replacing fine aggregates in concrete is between 20% and 40% [16,17,18,19,20]. Additionally, the physical properties and pozzolanic activity of iron tailings can improve the durability of concrete. Numerous researchers have discovered that incorporating iron tailings in concrete can reduce its drying shrinkage [21,22,23] and bolster its resistance to chloride ion corrosion, acid attack, and carbonation [24,25,26,27]. These research findings present potential opportunities for applying iron tailings in the realm of concrete. 

Currently, there is a noticeable research gap concerning the changes in mechanical properties and failure mechanisms of ITC under complex environmental factors. Iron tailings are widely distributed in the eastern region of China. Whether ITC can be widely used depends on whether it can withstand the influence of climate and environment in the eastern region. As a common environmental factor in the eastern region of China, the dry–wet cycle and negative temperature effect have an important impact on the performance and structure of iron tailings concrete. During these cycles, iron tailings concrete is exposed to fluctuating humidity and temperature conditions, potentially resulting in water penetration, chemical reactions, and other effects that lead to intricate changes in its properties. This study examines the mechanical properties of iron tailings concrete with varying river sand substitution rates after dry–wet cycles and negative temperature, ultimately determining the optimal substitution rate of iron tailings for fine aggregate. By analyzing the water content and porosity of iron tailings concrete, the study investigates the enhancement mechanism of iron tailings on concrete’s mechanical properties. This provides valuable insights for optimizing the mix proportion of iron tailings concrete and improving the durability of concrete structures.

## 2. Materials and Experimental Methods

### 2.1. Materials

#### 2.1.1. Cementitious Materials

Type I ordinary Portland cement 42.5 R was used as cementitious material in this test. The basic physical properties of the cement are shown in Table 1.

#### 2.1.2. Aggregates

Fine aggregates: The iron tailings were mined from the LiZhu iron tailings repository area in Zhejiang Province, China. The fineness modulus of iron tailings is 1.29 and that of river sand is 2.49, respectively. The particle size distribution of iron tailings and river sand is shown in Figure 1. The basic physical properties of iron tailings and river sand are presented in Table 2. The mineral compositions of cement, iron tailings, and river sand are shown in Table 3.

Coarse aggregates: The coarse aggregate uses crushed stone with a continuous gradation of 5–16 mm in particle size. The physical properties of coarse aggregate are shown in Table 4.

### 2.2. Specimen Casting and Curing Conditions

The concrete mix proportions can be found in Table 5. The designed strength grade of the iron tailings sand concrete was C30 with a water-to-cement ratio of 0.44. Four groups of concrete specimens were prepared, including OPC (control group), ITC10, ITC20, and ITC30, where ITC refers to iron tailings sand replacement percentage of river sand, with replacement percentages of 0%, 10%, 20%, and 30%, respectively. Cubic specimens with dimensions of 400 mm × 100 mm × 100 mm and 100 mm^3^ were prepared for each group of concrete specimens.

### 2.3. Test Method

#### 2.3.1. The Dry–Wet Cycle and Negative Temperature Environment Tests of Concrete

Guided by the GBT50082-2009 [28], the test of dry–wet cycles was conducted by a rapid dry–wet machine. The solution selected was water and 3.5% NaCl. The dry–wet cycle test was carried out in cycles of 2 days, with 1 day of immersion and 1 day of drying. The negative temperature test involves placing the specimen in an ambient chamber at −10 °C for 20 days on ice. The coupling test is a negative temperature test followed by a dry–wet cycle test. The 100 mm × 100 mm × 400 mm specimens were prepared for testing relative dynamic elastic modulus and flexural strength after 0, 20, 40, 60, 80, and 100 dry–wet cycles in different solutions and after negative temperature. The 100 mm^3^ specimens were prepared for testing compressive strength, weight loss, and splitting tensile strength after 0, 20, 40, 60, 80, and 100 dry–wet cycles in different solutions and after negative temperature. The experimental process is shown in Figure 2.

#### 2.3.2. Mechanics Test

The compressive strength, splitting tensile strength, and flexural strength were tested in a 1000 kN MTS microcomputer-controlled electro-hydraulic servo universal testing machine by changing the chuck. The compressive strength and split tensile strength were investigated by testing 100 mm^3^ cubes. For the flexural strength, the 400 mm × 100 mm × 100 mm were used for testing, and the loading speed was selected at 0.001 mm/s, 0.01 mm/s, 0.1 mm/s, and 1 mm/s. In this case, the double-K criterion is used to calculate the cracking toughness and instability toughness [29]. The formula for calculating the fracture toughness of ITC according to the double-K fracture criterion is as follows:(1)$$K_{IC}^{Q}=1.9(F_{Q}+\frac{mg}{2}\times 10^{-3})$$
where $K_{IC}^{Q}$ is the cracking toughness, which determines the ability of the material to resist cracking under load (MPa·m^0.5^); *F_Q_* is the cracking load, which is determined from the load–crack opening displacement process curve (kN); *mg* is the self-weight of the specimen between effective spans (kN). Converted from the total mass S/L as
(2)$$K_{IC}^{S}=1.9(F_{\max}+\frac{mg}{2}\times 10^{-3})\sqrt{a_{c}}f(\alpha )$$
where $K_{IC}^{S}$ is the instability toughness (MPa·m^0.5^); *F*_max_ is the instability load (kN); *a*_c_ the effective crack length (mm).

#### 2.3.3. Low-Field Nuclear Magnetic Resonance (NMR)

Using the MacroMR12-100H-GS low-field NMR (nuclear magnetic resonance) spectrometer, test the NMR T_2_ spectra of the sample after vacuum pressure saturation. Additionally, by utilizing the test standard sample’s NMR T_2_ spectra, establish a calibration line and calculate the sample’s porosity. The formula for calculating the pore size of concrete can be expressed as follows:(3)$$\frac{1}{T_{2}}=\rho \left(\frac{S}{V}\right)$$
where *ρ* represents the transverse surface relaxation strength of porous media, in units of μm/ms, and its value varies depending on the sample; *S* represents the surface area of the pores; *V* is the volume of the pores. The *ρ* of concrete is generally 3–10 μm/ms. Based on experience, *ρ* is usually set to 5 μm/ms.

Assume the pores are perfect spheres:(4)$$S/V=3/r_{c}$$
where *r_c_* represents the radius of the pores. By substituting Equation (4) into Equation (3), the T2 relaxation time distribution image can be transformed into the pore size distribution image of concrete. This allows for an intuitive analysis of the pore size distribution in each test sample.

## 3. Results and Discussion

### 3.1. Analysis of the Mechanical Properties of ITC

The compressive strength and splitting tensile strength of the iron tailings concrete were tested after 28 days of standard curing, and the flexural strength was tested under different loading rates (0.001 mm/s, 0.01 mm/s, 0.1 mm/s, 1 mm/s). To minimize experimental error, six specimens were tested in each group. The average value was then calculated and used as the experimental result, which is illustrated in Figure 3. Among these tests, the concrete with a 20% iron tailings replacement (ITC20) demonstrated the best mechanical performance, with increases in compressive strength and splitting tensile strength of 3.14 MPa and 0.49 MPa, respectively, compared to the control group. This result was in agreement with Li and Zhang [17,30]. As Figure 3c shows, the flexural strength of the concrete was observed to be highest at a loading rate of 1 mm/s, which can be attributed to the concrete’s sensitivity to loading rates. Its flexural strength generally increases with higher loading rates [31]. Moreover, regardless of the loading rate used in the test, the flexural strength of the concrete showed a trend of initially increasing and then decreasing with increasing iron tailings content. Among them, the concrete with 20% iron tailings replacement (ITC20) demonstrated the best performance, with an increase in flexural strength of 11.68%, 11.69%, 15.79%, and 8.41% compared to the control group at different loading rates. The mechanism through which iron tailings enhance the mechanical properties of concrete is primarily attributed to their distinct physical and chemical properties. Iron tailings, as industrial waste, possess a small particle size and a low fineness modulus (only 51.81% of that of river sand). Incorporating them into the mix can rectify the grading deficiencies in concrete, fill its internal voids, and increase its compactness. Moreover, iron tailings particles exhibit rough surfaces, a large specific surface area, and high hardness, which allow them to bond securely within the concrete. This enhances the overall integrity of the concrete and hinders the formation and propagation of cracks during flexural strength tests.

### 3.2. Analysis of ITC Pore Structure

Using hydrogen atomic nuclei as the detection target, we have drawn the T2 spectral test curve (Figure 4) and the pore size distribution diagram (Figure 5) of ITC concrete through low-field nuclear magnetic resonance experiments. In the T2 spectral test curve, void size is directly proportional to the corresponding peak relaxation time. The larger the relaxation time, the larger the pore size; similarly, a larger peak area corresponds to a greater number of voids. Analyzing the pore size distribution diagram of concrete (Figure 5) revealed that the concrete with a 20% iron tailings replacement for river sand exhibited the smallest pores. This suggests that the pore size distribution of concrete, when fine aggregates are replaced with iron tailings, predominantly consists of small pores. Moreover, the quantity of larger pores in the ITC20 is significantly less than in ordinary concrete specimens, which is consistent with the regularities reflected in the T2 spectrum. This further validates that replacing a certain amount of fine aggregate sand with iron tailings can increase the compactness of the concrete specimens and optimize their pore structure, thereby enhancing the mechanical properties of the specimens. Additionally, the pore size distribution of ITC30 is better than that of OPC and ITC10, indicating that within the range of a 20–30% substitution rate, iron tailings make a significant contribution to the grading adjustment of fine aggregates. However, due to the low replacement rate in ITC10, the fine aggregate sand in the specimen did not reach the optimal grading state during preparation, and the grading distribution was changed from the non-replaced situation, resulting in an increased distribution of large pore size in certain areas.

After calibrating the standard specimens and normalizing their volumes, the calculated moisture content and porosity of the four groups of concrete specimens are shown in Table 6. The data in Table 6 suggest that the moisture content and porosity of the concrete initially increase and then decrease with rising iron tailings content. When the replacement rate of river sand with iron tailings is at 20%, the moisture content and porosity of the concrete specimens reach their lowest values. Specifically, the moisture content is 3.78 g less than that of OPC, and the porosity is 3.27% lower than that of OPC. This can be attributed to the small particle size of iron tailings, which optimizes the pore structure within the concrete and enhances its overall integrity.

### 3.3. Effect of Dry–Wet Cycles on Mechanical Properties and Durability of ITC

#### 3.3.1. Weight Loss

The weight loss data of specimens subjected to dry–wet cycles were measured and presented in Figure 6. In the figure, negative values indicate an increase in mass, and positive values indicate a decrease in mass, based on the reference group of OPC concrete. A comparison of the quality of concrete specimens after standard curing for 28 days (with 0 dry–wet cycles) revealed that the quality of ITC was superior to that of OPC. In particular, ITC20 showed a significant 16% increase in mass compared to OPC. The improved density and quality of iron tailings concrete can be attributed to the appropriate addition of iron tailings, which optimizes the internal pore structure of the concrete and increases its compactness.

The mass of specimens in each group exhibited a trend of initially increasing and then decreasing with an increase in the number of dry–wet cycles when a water solution was used as the medium for dry–wet cycle testing. After 20 dry–wet cycles, the mass of OPC, ITC10, ITC20, and ITC30 increased by 4%, 6%, 7%, and 3%, respectively. The increase in concrete mass during the first 20 cycles of dry–wet conditions can be attributed to the reaction of unhydrated clinker in the concrete with water, producing more cementitious materials. The moisture penetrated further into the interior of the concrete as the number of dry–wet cycles increased, leading to the generation and development of micro-cracks and the peeling of aggregates due to water erosion, resulting in weight loss of the concrete specimens. After 100 dry–wet cycles, OPC experienced a weight loss of 39%, while ITC20 exhibited a weight loss of only 34%, demonstrating the best performance. This suggests that the appropriate addition of iron tailings helps to enhance the concrete’s resistance to erosion by the water solution and strengthen the overall integrity of the concrete.

The weight loss rate and corresponding apparent changes in concrete after chloride salt dry–wet cycles are shown in Figure 6b and Figure 6c, respectively. As seen in Figure 6b, the concrete is severely affected by chloride salt erosion, with a continuous decrease in mass as the number of dry–wet cycles increases. The degradation of the concrete and the weight loss rate significantly increase, particularly after 40 cycles. After 100 dry–wet cycles, the weight loss rates of OPC, ITC10, ITC20, and ITC30 are 55%, 52%, 47%, and 59%, respectively. Among them, ITC20 has the lowest weight loss rate, and it can be observed from the apparent changes in the specimens that ITC20 has the lightest deterioration on the surface. However, varying degrees of disintegration and shedding were observed at the edges and corners of ITC concrete specimens. Scaling and degradation of the concrete are related to the infiltration of chloride ions. In a wet environment, chloride ions can permeate into the micro-cracks of the concrete and react with the Ca^2+^ in the C-S-H gel to form chlorides. When the environment becomes dry, these chlorides crystallize within the concrete, leading to the formation of cracks and peeling on the concrete surface. Concurrently, the precipitated chlorides form crystals on the surface of the concrete, affecting its appearance. Additionally, with the reduction of C-S-H gel, the mechanical properties and durability of the concrete begin to decline [32]. A visible manifestation of this is the peeling of aggregates on the surface of the concrete specimens. Previous studies have shown that the distribution pattern of chloride ion concentration in concrete is highest at the edges and gradually decreases toward the center. Therefore, the phenomenon of spalling and detachment at the edges of ITC specimens after 100 dry–wet cycles (NaCl solution) can be explained by the improvement of compactness of concrete by the iron tailings, which enhances its resistance to chloride corrosion. However, with the accumulation of time, the continuous increase in chloride concentration inside the concrete leads to a decrease in the C-S-H gel, which reduces the adhesive capacity between the aggregates. This negative effect is particularly significant for iron tailings with small particle sizes and large specific surface areas. These experimental results indicate that if the dry–wet cycles (NaCl) continue, a sudden change in the degradation of the ITC surface may occur.

#### 3.3.2. Relative Dynamic Elastic Modulus

The magnitude of the relative dynamic elastic modulus reflects the ability of a material to resist elastic deformation, with higher values indicating a smaller likelihood of elastic deformation occurring. In this article, we present the variation of the relative dynamic elastic modulus of concrete after undergoing dry–wet cycles (Figure 7), which is similar to the mass loss rate of concrete under the same conditions. Figure 7a reveals two main trends in the change in the relative dynamic elastic modulus of concrete under the dry–wet cycles in a water-soluble solution: (1) As the number of dry–wet cycles increases, the relative dynamic modulus of the concrete first increases and then decreases, reaching its maximum value at 20 cycles; (2) As the amount of iron tailings added increases, the relative dynamic elastic modulus of the concrete also first increases and then decreases. Among all test groups, ITC20 exhibited the best performance, with a relative dynamic elastic modulus of 43.65 GPa, which is 5.55 GPa higher than that of OPC concrete. After 20 dry–wet cycles, the relative dynamic elastic modulus of ITC20 increased to 45.71 GPa. After 100 dry–wet cycles, the relative dynamic elastic modulus of ITC20 decreased to 32.57 GPa, a reduction of 28.71%, while the control group’s relative elastic modulus decreased by 34.09%. In the early stage of the dry–wet cycle, the increase in the dynamic modulus of elasticity of concrete may be due to the continued hydration reaction of unreacted components such as C_3_S in cement after the concrete absorbs water, resulting in the formation of more C-S-H gel. This process reduces porosity and optimizes the microstructure of the concrete, thereby enhancing its dynamic modulus of elasticity [33]. Subsequently, as hydration intensifies, the concrete will generate some expansive complex salts (Ca(OH)_2_·H_2_O crystals and magnesium salt crystals). These complex salts do not grow sufficiently in the initial stage, filling the concrete internally and improving its compactness. However, as the dry–wet cycles continue, the expansive stress generated by the complex salt crystals increases. When the expansive force exceeds the adhesive force between the aggregates and the mortar, the concrete begins to form damage, and the relative dynamic elastic modulus starts to decrease. Iron tailings concrete, due to its higher compactness and weaker erosion by the water-soluble solution, slows down the damage rate.

Figure 7b indicates that the relative dynamic elastic modulus of concrete continuously decreases during the dry–wet cycles in a chloride salt environment. This is because concrete will generate more and faster expansive complex salts in a chloride salt environment (in addition to the aforementioned Ca(OH)_2_·H_2_O and magnesium salt crystals, it also includes complex salts such as CaCl·Ca(OH)_2_·nH_2_O) [32]. This leads to a sharp increase in the internal expansive stress of the concrete, exceeding the adhesive force between the aggregate and the cementitious material, causing internal damage to the concrete and the relative dynamic elastic modulus to begin to decrease.

#### 3.3.3. Mechanical Property

The compressive strength and splitting strength of concrete after dry–wet cycles are shown in Figure 8. As can be seen from Figure 8a,b, the change in mechanical properties of OPC concrete and ITC concrete in pure water dry–wet cycles is consistent with mass loss and relative dynamic elastic modulus. In the early stage of the test, the mechanical performance increases and begins to decrease after 20 dry–wet cycles. The reason for this is that water absorption in the early stage promotes the reaction of unhydrated clinker, generating more gel and improving the mechanical performance of the concrete. However, as the number of cycles increases, the concrete’s inherent strength is insufficient to resist the internal tensile stress, leading to the formation of micro-cracks and the beginning of concrete deterioration. The formation of cracks provides a pathway for the intrusion of water-soluble solutions. Therefore, during the 60–100 dry–wet cycles, the deterioration of iron tailings concrete accumulates and deepens due to the previous stage. In this stage, the mechanical performance of both OPC and ITC concrete declines significantly. However, the decrease in the mechanical performance of ITC concrete is relatively more moderate, with ITC-20 having the smallest decline, the least deterioration, and the least affected by the dry–wet cycles.

From Figure 8c,d, it can be found that after chloride salt dry–wet cycles, the mechanical performance of ITC20 remains the best. After 40 dry–wet cycles in chloride salt, the compressive strength and splitting strength of concrete specimens begin to decline significantly. This is because, as the number of cycles increases, the expansive stress generated by CaCl·Ca(OH)_2_·nH_2_O complex salt formed by the reaction of chloride salt with cement rapidly increases, leading to the formation of micro-cracks in the concrete. On one hand, this causes specimen deterioration, and on the other hand, it promotes further intrusion of chloride ions in subsequent cycles. Furthermore, the infiltrating chloride ions react chemically with the cement hydration products, destroying the balance between Ca(OH)_2_ and C-S-H gel, causing the C-S-H gel to decompose and lose its binding ability [33].

### 3.4. Effect of Negative Temperature on Mechanical Properties of ITC

The relationship between the cracking resistance of ITC under negative temperature action, the substitution rate, and the loading rate is depicted in Figure 9. From Figure 9, it can be inferred that the starting fracture toughness and destabilization fracture toughness of ITC, with a 10–20% iron tailings replacement rate, outperform those of standard concrete specimens. This suggests that the integration of iron tailings can effectively enhance the fracture resistance of concrete. This improvement occurs because the CaO and Al_2_O_3_ components in the iron tailings react chemically with water to produce alkaline substances during concrete formation. This reaction promotes cement hydration during hardening, leading to more complete hydration. Consequently, this process significantly reduces the initial defects in concrete during hardening, resulting in a notable increase in fracture toughness and instability toughness at a macro-scale [34]. Figure 9b,d both demonstrate a significant upward and subsequent downward trend, with the inflection point at 20%, indicating that the most beneficial effect on the hardening of concrete and its crack resistance is achieved at a 20% replacement dose. At the same time, the cracking resistance at negative temperatures is enhanced, with a rise of 5.56% and 10.66% in crack initiation toughness and instability toughness at 0.1 mm/s and 20% IT, respectively. This is due to the ability to control the temperature during its hydration and the temperature-derived microcracking at negative external temperatures, which manifests itself as an increase in its crack resistance at negative temperatures. In addition, it can also be seen from the graph that the pattern of influence of different loading rates on the cracking toughness and destabilization toughness of plain concrete and ITC is almost the same, regardless of whether the specimens are at room temperature or at a negative temperature. The overall finding is that the cracking resistance of ITC surpasses that of ordinary concrete when the loading rate ranges from 0.001 mm/s to 10 mm/s. The fracture damage at lower loading rates is primarily plastic, while at higher loading rates, the damage is predominantly brittle [35]

In conclusion, the replacement doping demonstrates a more pronounced effect on the fracture toughness and instability fracture toughness at negative temperatures. It is suggested that the optimum replacement admixture level for recycled concrete using this type of iron tailings sand to replace conventional sand should be controlled at around 20% to show optimum crack resistance, while the loading rate is optimum at 0.1 mm/s.

### 3.5. Effect of Negative Temperature and Dry–Wet Cycles on Mechanical Properties of ITC

Based on the results of the mechanical properties tests in a single environment, 20% iron tailings was initially determined as the optimum admixture, and the mechanical properties were investigated under the combined effect of negative temperature and dry–wet cycles (Figure 10). The results demonstrate that the mechanical properties of ITC20 in the coupled environment are better than those of ordinary concrete. In the first 40 cycles, the mechanical properties of plain concrete and ITC gradually decreased as the number of cycles increased, although the decrease was not significant (within 10%). After 40 cycles, the mechanical properties of plain concrete and ITC continued to decline. In contrast, the mechanical properties of ITC20 consistently outperformed those of plain concrete in both the water and sodium chloride solution cycles. This is consistent with the pattern of change in mechanical properties under the influence of a single factor, indicating that 20% of IT has a higher capacity to resist cracking and maintain stability under load than OPC. The active minerals in the iron tailings facilitate an increase in hydration products, contributing to an enhanced bond between the coarse aggregate and the cement slurry. This ultimately slows down the deterioration of ITC in complex service environments.

## 4. Conclusions

In this paper, a comparative analysis of the mechanical properties of iron tailings concrete at room temperature, under negative temperature, and in a dry–wet cycle environment was conducted. We found that the research method of using tailings waste, which both pollutes the environment and occupies land resources, to quantitatively replace non-renewable fine aggregate sand to manufacture recycled concrete is feasible and made a preliminary attempt for the research and application of iron tailings sand recycled concrete in the monsoon frost zone. The main findings from this research can be summarized as follows:The optimal replacement rate of iron tailings for river sand is 20%, at which point the concrete exhibits the smallest porosity and the best mechanical performance.The incorporation of iron tailings increases the water demand of concrete, which promotes the hydration reaction of cement, resulting in increases in mass, relative dynamic elastic modulus, and mechanical performance during the early stages of the dry–wet cycle tests.At a single factor of negative temperature, the best crack resistance was achieved at a 20% iron tailings replacement admixture rate. This crack resistance was higher than that of the same concrete material at room temperature. Moreover, the greatest fracture toughness value was achieved at a controlled loading rate of 0.1 mm/s.In a combined negative temperature and dry–wet cycle environment, the mechanical properties of concrete with a 20% iron tailings replacement rate surpass those of standard concrete.

## Figures and Tables

**Figure 1 materials-16-04602-f001:**
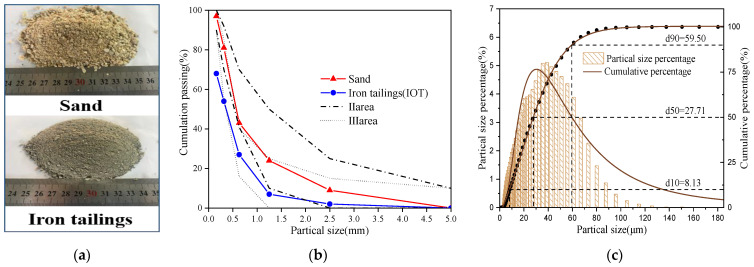
(**a**) Material appearance, (**b**) granular gradation of IT and sand; (**c**) particle size distribution of iron tailings.

**Figure 2 materials-16-04602-f002:**
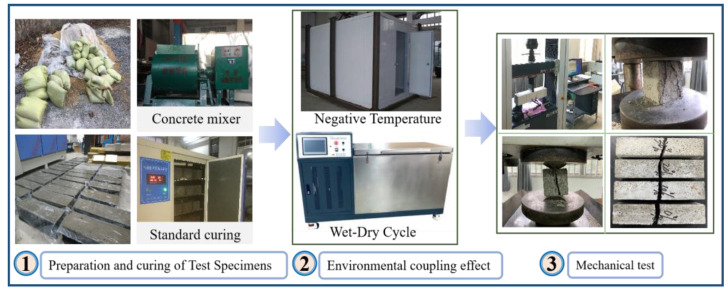
Flow chart of concrete specimen preparation and mechanical test.

**Figure 3 materials-16-04602-f003:**
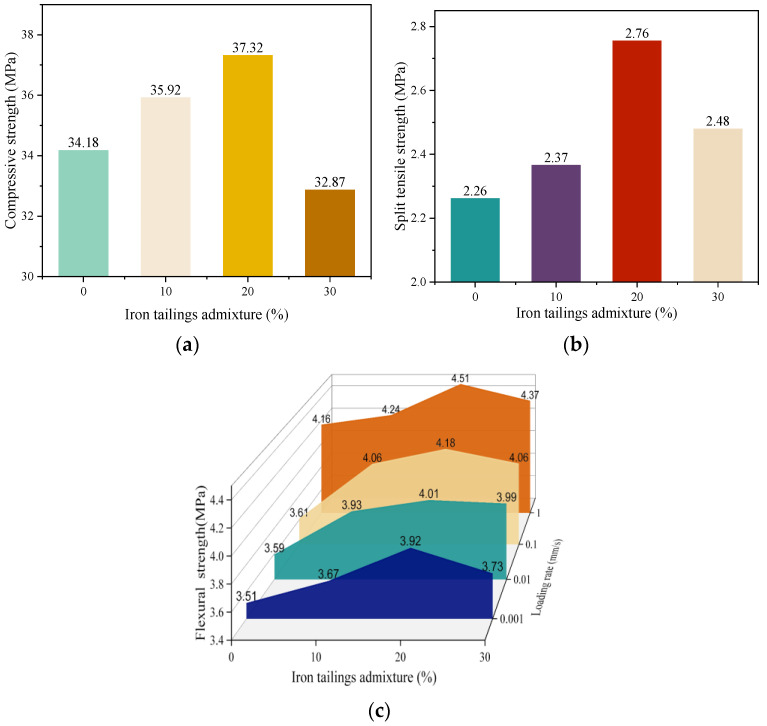
Mechanical performance of ITC cured for 28 d. (**a**) Compressive strength. (**b**) Splitting tensile strength. (**c**) Flexural strength.

**Figure 4 materials-16-04602-f004:**
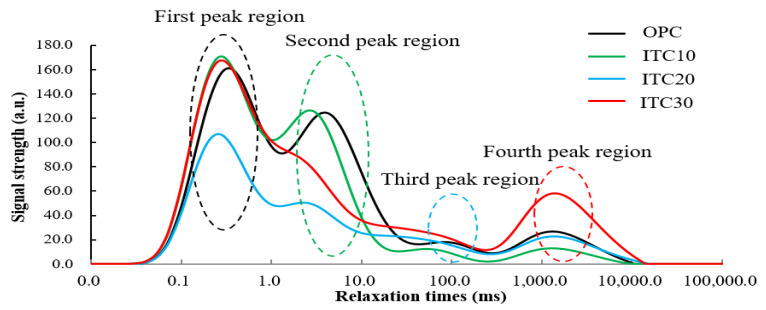
T2 spectrum test curve of ITC cured for 28 d.

**Figure 5 materials-16-04602-f005:**
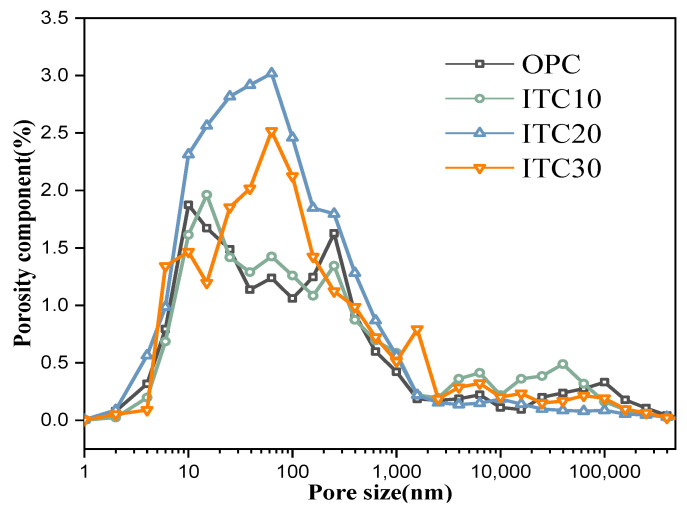
Pore size distribution curve of ITC20.

**Figure 6 materials-16-04602-f006:**
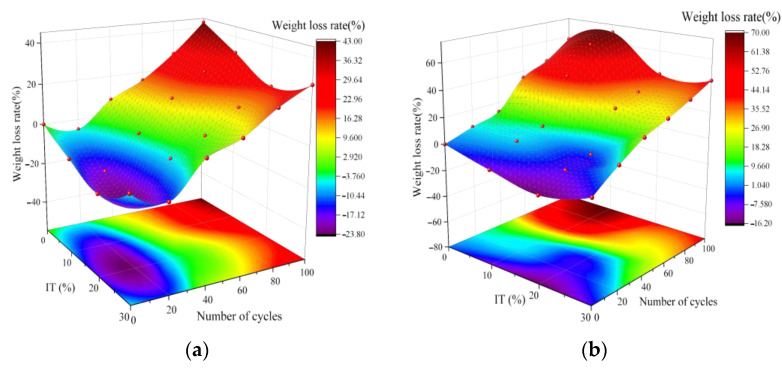
Weight loss rate of ITC under dry–wetting cycle. (**a**) Water solution. (**b**) NaCl solution. (**c**) Apparent changes.

**Figure 7 materials-16-04602-f007:**
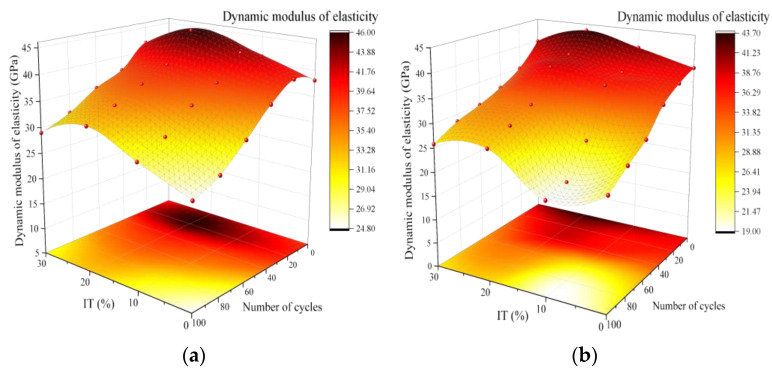
Relative dynamic elastic modulus of ITC under dry–wetting cycle. (**a**) Water solution. (**b**) NaCl solution.

**Figure 8 materials-16-04602-f008:**
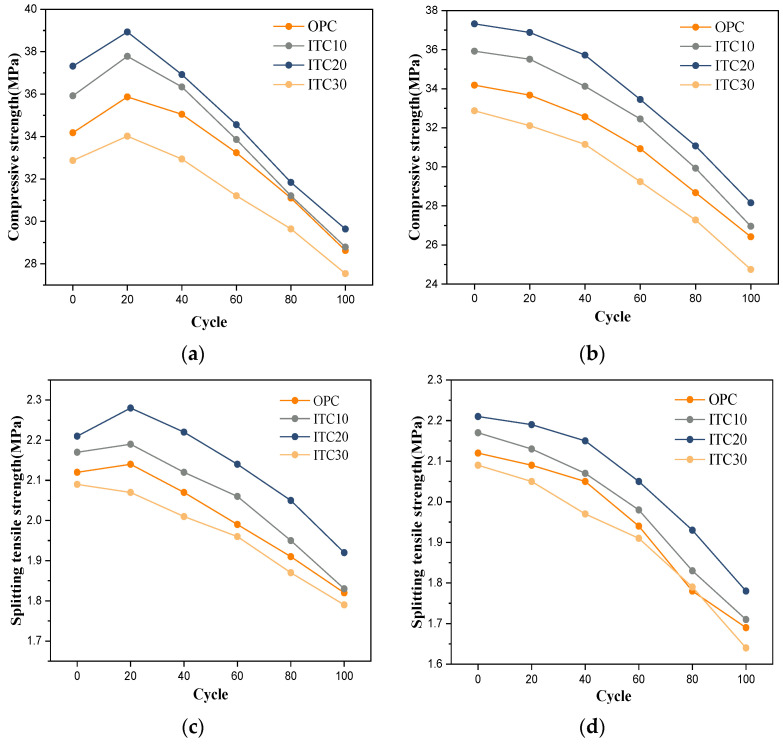
Compressive strength and splitting tensile of ITC under dry–wetting cycle: (**a**,**c**) in water, (**b**,**d**) and in NaCl solution.

**Figure 9 materials-16-04602-f009:**
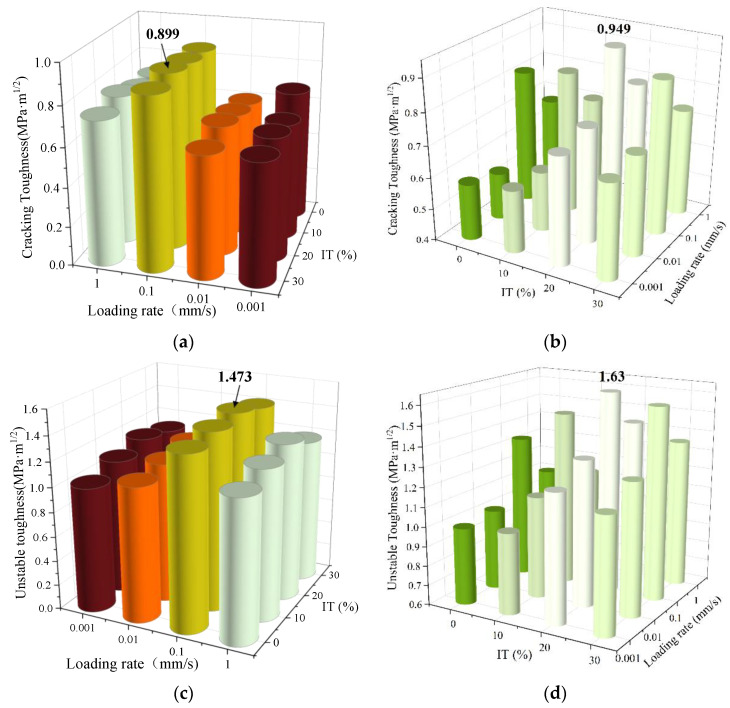
Crack initiation toughness and unstable toughness of ITC before and after negative temperature. (**a**) Crack initiation toughness before negative temperature. (**b**) Crack initiation toughness after negative temperature. (**c**) Unstable toughness before negative temperature. (**d**) Unstable toughness after negative temperature.

**Figure 10 materials-16-04602-f010:**
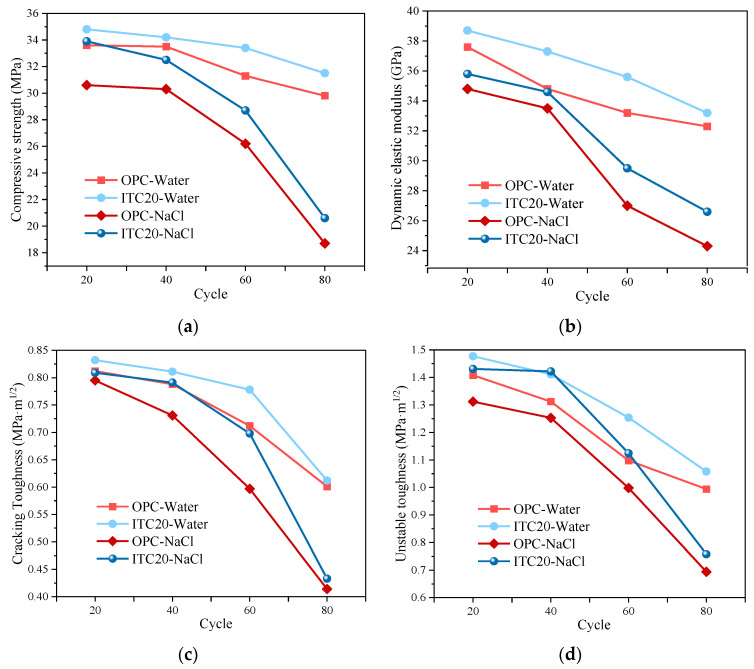
Mechanical properties of ITC under negative temperature and dry–wet cycles. (**a**) Compressive strength. (**b**) Dynamic modulus. (**c**) Crack initiation toughness. (**d**) Unstable toughness.

**Table 1 materials-16-04602-t001:** Characteristics of Portland cement.

Flexural Strength (MPa)		Compressive Strength (MPa)		Specific Surface Area (g/cm^2^)	Setting Time (min)	
3 days	28 days	3 days	28 days	367	Initial setting	Final setting
4.1	9.3	21.2	46.1		164	272

**Table 2 materials-16-04602-t002:** Characteristics of iron tailings and sand.

Physical Properties	Apparent Density (kg/m^3^)	Fineness Modulus (MX)	Type	Water Absorption (%)	PH Value
Sand	2.620	2.49	Medium sand	1.67	8
Iron tailings	2.855	1.29	Fine sand	3.65	10

**Table 3 materials-16-04602-t003:** Chemical properties of Portland cement, iron tailing, and sand (wt%).

Element	MgO	Fe_2_O_3_	Al_2_O_3_	SiO_2_	CaO	K_2_O	Other
Cement	1.40	4.04	5.80	21.47	59.64	-	4.67
Iron tailings	16.7	10.80	8.19	41.11	13.99	0.72	0.75
Sand	-	-	-	99.8	-	-	0.2

**Table 4 materials-16-04602-t004:** Physical properties of coarse aggregates.

Coarse Aggregates	Particle Size (mm)	Apparent Density (kg/m^3^)	Water Absorption (%)	Crushing Index (%)
Natural Crushed Stone	5–16	2530	1.17	13

**Table 5 materials-16-04602-t005:** Mixing proportions of concrete.

Sample Number	Water (kg/m^3^)	Cement (kg/m^3^)	Sand (kg/m^3^)	Iron Tailing (kg/m^3^)	Stone (kg/m^3^)	W/C	Slump (mm)
OPC	180	409	535	0	1249	0.44	200
ITC-10%	180	409	481	54	1249	0.44	196
ITC-20%	180	409	427	108	1249	0.44	191
ITC-30%	180	409	378	162	1249	0.44	183

**Table 6 materials-16-04602-t006:** NMR characteristic parameter calculation results.

Sample	V/cm^3^	NMR Signal Intensity (a.u.)	Water Content (g)	Porosity (%)
OPC	60.76	4193.56	9.92	12.83
ITC10	63.89	3806.29	9.28	11.36
ITC20	58.16	2520.14	6.14	9.56
ITC30	61.68	4089.27	10.48	13.67

## Data Availability

Subsequent articles being written will be based on existing data, so it is not convenient to provide raw data at this time. If you would like to know about our trial process and data sources, please contact our email address, we will share without reservation.

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
