# Peer review of "Investigation of the Mechanical Properties of Iron Tailings Concrete Subjected to Dry–Wet Cycle and Negative Temperature"

_materials, 2023, doi:10.3390/ma16134602_

Round 1
Reviewer 1 Report
This paper discusses the mechanical properties of iron tailings concrete with varying river sand substitution rates after wet-dry cycles and negative temperature, ultimately determining the optimal substitution rate of iron tailings for fine aggregate. The topic is interesting and warrants a constructive discussion on the utilization of iron tailings in concrete. However, several points as indicated below need to be addressed by authors to improve the quality of the articles.
p.2 Line9
"ITC" is the first one here. ” ITC (Iron Tailings Concrete)” should be explained here.
p.2 Line17
“…in concrete being between 20% and 40% [16-20]”
Is it the amount added to fine or coarse aggregate? Please explain in detail.
p.3 Table1
What is “Fineness”? And please indicate the specific surface area of the cement.
p.3 Table2
In general, the water absorption of fine aggregates for concrete is less than 3%. Is the water absorption of river sand the correct value? And please also indicate the physical properties of the coarse aggregate.
p.3 Table4
Please indicate the unit quantity of ITC. Also, what is the sand ratio?
p.3 “2.2 Specimen casting and curing conditions”
I think that the iron tailings will adversely affect the fluidity of the concrete.
Please indicate the slump and air content of the concrete after mixing. This information is essential for further discussions regarding strength and durability.
p.4 “2.3 Test method”
Please describe the temperature conditions for the dry-wet cycle test.
The relationship between the dry-wet cycle test and the negative temperature test is unclear. It would be helpful to provide a diagram illustrating the test procedure for each test after curing.
p.5 “3.3 Analysis of ITC pore structure of ITC”
Details of the low-field nuclear magnetic resonance experiments are unknown. Please add to the description of the test method.
p.6 Table 5
Please provide the calculation method and basis for the calculation.
p.6 Figure 5
The pore size distribution of all mixes should be presented and discussed.
p.7 Line 1
Please explain the moisture condition during the weight loss measurement.
p.7 Line10 from the bottom
Please include the reference numbers of your previous studies.
p.8 Figure6(a)(b)
“Quality loss ratio” replace by “Weight loss rate”.
p.9 Line12-20, p.10 Line11-18, p.11 Line1-6
Chemical discussions have been considered, but there are no data available in this experiment to support this. If the discussion is based on previous findings or other references, please provide the relevant references.
Reviewer 2 Report
The article deals with an experimental and analytical study about the mechanical and durability properties of iron tailings concrete subjected to wet-dry cycle and negative temperature. Low-field nuclear magnetic resonance is employed to evaluate the pore characteristics of the material as interesting technology.
Currently, it is an important topic in concrete technology the use of industrial solid waste to replace aggregates because. The article is well written and includes an interesting experimental analysis, complete and easy to follow for the reader.
The reviewer has noted some things to correct and discuss throughout the paper (pdf document, in yellow color with notes). Moreover, the reviewer recommends an effort to include and support the analysis and results with more scientific references.
The article is relevant to the readers of Materials, and the reviewer recommends its acceptance for publication once the suggestions (included in pdf document) are solved and only if the scientific discussion is supported with scientific references.

Reviewer 3 Report
General Comment
The submitted manuscript presents an experimental study on the mechanical properties and durability of iron tailings concrete (ITC) subjected to wet-dry cycles and negative temperature. For this, concrete specimens were fabricated in the laboratory, the main variable study was the percentage of iron tailing sand replacement. After a literature review, and after identifying the research gap in this field, the manuscript presents the raw materials and describes the used experimental methods. The results are presented and discussed, namely: the compressive and splitting strength for different loading rates, the ITC pore structure, and the effect of dry-wet cycles (both in water and NaCl solution) and negative temperature on the mechanical properties and durability of ITC. From the obtained results, the authors conclude about the feasibility to replace fine aggregate sand to manufacture recycled concrete, and found an optimum replacement of 20%.
The topic of the manuscript is very interesting and important because it deals with recycled concrete incorporating iron tailings which constitutes a by-product of mining operations which poses several threats for ecological environment. The main novelty of the study are the changes in mechanical properties and durability ot ITC under complex environmental factors. This study constitutes a good contribution to the field and the results can be useful for futures studies and also for concrete suppliers.
I consider that the manuscript requires a revision before it can be accepted for publication. I made some comments in order to improve the manuscript. The authors should take the comments into account and revise their manuscript.
Specific Comment 1
The manuscript needs a revision to correct typos and improve the reading throughout the text. Just some examples:
- page 2, ITC should be defined the first time it appears in the text;
- Figure 2: correct “Paretical” to “Particle” inside the graph;
- For instance, correct “3.14MPa” to “3.14 MPa” (idem for all similar situations);
- Rephrase title of section 3.2 (ITC is repeated twice);
- …
Specific Comment 2
The abstract should summarize the found benefices of incorporating IT in concrete, compared with the OPC. In addition, keywords should also incorporate “Iron tailing concrete”.
Specific Comment 3
Section 2
Please add a photo of the used IT as raw material.
Also, please add photos of all the testing procedures, including the ITC samples in testing position to obtain the mechanical properties (before and after failure).
Specific Comment 4
In Table 4, it seems that units are missing.
Specific Comment 5
Section 3
Whenever possible, please briefly discuss how your results compare with the ones from previous related studies from the literature review. This would add value to your manuscript.
Specific Comment 6
Figure 3
The graphs in Figure 3(a) and 3(b) present large bars errors.
How many samples were tested to compute the error?
Also, please give a possible explanation for the presented large bar errors, and how this can impact your discussion and results. Please note that, for Figure 3(b), the minimum value obtained for the split tensile strength of all ITC is almost the same. How can you be sure that the ITC with 20% is better? This must me well justified in the manuscript.
Specific Comment 7
Section 3.2
Please present details on how the moisture content and porosity were computed.
The English Language is fine, minor editing is required.
Round 2
Reviewer 1 Report
Thank you for your appropriate response to the peer review comments. I think this manuscript will be acceptable. However, please correct the following points.
p.3 Table1
Please verify the accuracy of the specific surface value and indicate the required units(g/cm2).
p.4 Table5
The unit of the unit content value should be corrected to kg/m3.
Reviewer 3 Report
The authors have improved the manuscript according to all my previous comments. I consider that the manuscript can be accepted for publication after a minor revision to correct some typos and formatting issues.
Some examples:
- Correct "Mpa" to "MPa";
- Correct "mm^3";
- Leave a space between the number and the unit;
...
Minor editing of English language required
Author Response
Thank you for your comments, please see the attachment.
